# Highly Efficient Bioconversion of *trans*-Resveratrol to *δ*-Viniferin Using Conditioned Medium of Grapevine Callus Suspension Cultures

**DOI:** 10.3390/ijms23084403

**Published:** 2022-04-15

**Authors:** Su Hyun Park, Yu Jeong Jeong, Sung-Chul Park, Soyoung Kim, Yong-Goo Kim, Gilok Shin, Hyung Jae Jeong, Young Bae Ryu, Jiyoung Lee, Ok Ran Lee, Jae Cheol Jeong, Cha Young Kim

**Affiliations:** 1Biological Resource Center, Korea Research Institute of Bioscience and Biotechnology (KRIBB), Jeongeup 56212, Korea; suhyun@kribb.re.kr (S.H.P.); yjjeong@kribb.re.kr (Y.J.J.); heypsc@kribb.re.kr (S.-C.P.); cherish767@kribb.re.kr (S.K.); kimyg@kribb.re.kr (Y.-G.K.); goshin@kribb.re.kr (G.S.); jiyoung1@kribb.re.kr (J.L.); 2Department of Applied Plant Science, College of Agriculture and Life Science, Chonnam National University, Gwangju 61186, Korea; mpizlee@jnu.ac.kr; 3Functional Biomaterial Research Center, Korea Research Institute of Bioscience and Biotechnology (KRIBB), Jeongeup 56212, Korea; hjeong21@kribb.re.kr (H.J.J.); ybryu@kribb.re.kr (Y.B.R.)

**Keywords:** conditioned medium, callus cultures, viniferin, resveratrol, bioconversion

## Abstract

*δ*-Viniferin is a resveratrol dimer that possesses potent antioxidant properties and has attracted attention as an ingredient for cosmetic and nutraceutical products. Enzymatic bioconversion and plant callus and cell suspension cultures can be used to produce stilbenes such as resveratrol and viniferin. Here, *δ*-viniferin was produced by bioconversion from *trans*-resveratrol using conditioned medium (CM) of grapevine (*Vitis labruscana*) callus suspension cultures. The CM converted *trans*-resveratrol to *δ*-viniferin immediately after addition of hydrogen peroxide (H_2_O_2_). Peroxidase activity and bioconversion efficiency in CM increased with increasing culture time. Optimized *δ*-viniferin production conditions were determined regarding H_2_O_2_ concentration, incubation time, temperature, and pH. Maximum bioconversion efficiency reached 64% under the optimized conditions (pH 6.0, 60 °C, 30 min incubation time, 6.8 mM H_2_O_2_). In addition, in vitro bioconversion of *trans*-resveratrol was investigated using CM of different callus suspension cultures, showing that addition of *trans*-resveratrol and H_2_O_2_ to the CM led to production of *δ*-viniferin via extracellular peroxidase-mediated oxidative coupling of two molecules of *trans*-resveratrol. We thus propose a simple and low-cost method of *δ*-viniferin production from *trans*-resveratrol using CM of plant callus suspension cultures, which may constitute an alternative approach for in vitro bioconversion of valuable molecules.

## 1. Introduction

Stilbenes are phenolic compounds produced by pants of various families, including Vitaceae, Pinaceae, and Leguminosae, and their synthesis is enhanced in response to various biotic and abiotic stressors. Resveratrol is one of the most extensively studied stilbene compounds because of its biological activities such as antitumor, neuroprotective, and antioxidant effects [1,2,3]. In addition to resveratrol, its derivatives such as viniferin, piceid, pterostilbene, and arachidin can exert biological functions which are beneficial to health [4]. *δ*-Viniferin is known as resveratrol dehydrodimer or an isomer of *ε*-viniferin, and this compound can exert antioxidant, anti-diabetic, and anti-diarrheal effects [5,6,7]. *trans*-Resveratrol and *δ*-viniferin occur naturally in red wine, grape canes, and grapevine leaves [8,9], and they are accumulated in grape leaves or grape berries following pathogen infection or abiotic stress such as ozone stress, ultrasonication, and UV radiation exposure [10,11,12]. Recently, it was shown that adding elicitors to grapevine cell cultures can increase the production of *trans*-resveratrol and *δ*-viniferin [13]. For example, Santamaria et al. [14] demonstrated that *δ*-viniferin production was enhanced in grape cell cultures after treatment with methyl jasmonate and jasmonic acid. We previously observed that addition of methyl jasmonate and stevioside to the culture medium of grapevine cell suspension cultures induced extracellular production of *δ*-viniferin, *ε*-viniferin, and *trans*-resveratrol [15]. Apart from bio-production of resveratrol derivatives, viniferin and resveratrol are currently produced by extraction from plant materials and chemical synthesis, and Takaya et al. [16] demonstrated the conversion of resveratrol to *ε*-viniferin by treatment with thallium(III) nitrate as an oxidizing inorganic reagent.

Bioconversion, also known as biotransformation, refers to the specific modification of a compound through a biocatalyst such as live microorganisms or enzymes [17]. In recent decades, bioconversion has attracted increasing interest due to its application in various fields such as the production of fuels, chemicals, and other commercial products [18]. In contrast to traditional chemical synthesis methods, bioconversion catalyzed by isolated enzymes occurs at near-neutral pH, atmospheric pressure, and ambient temperature. For this reason, bioconversion approaches are considered more eco-friendly regarding the production of organic compounds compared to chemical synthesis, which requires large amounts of organic solvents. Some previous studies reported that plant peroxidases or fungal laccases catalyze dimerization of resveratrol into *δ*-viniferin [19,20,21]. Moreover, human cyclooxygenase-1 can catalyze resveratrol dimerization to produce viniferin [22]. Enzymatic bioconversion is an eco-friendly alternative to chemical synthesis, however, isolation of enzymes from cells and purification of large amounts of commercially useable enzymes is very time-consuming.

Conditioned medium (CM) is defined as supernatant medium separated from cells in callus and cell suspension cultures, and CM contains extracellular proteins and metabolites that are secreted from the cells into the medium [23]. Such cultures constitute a useful system to study proteins secreted under biotic and abiotic stress conditions. CM thus not only represents a valuable source of information on molecular level cell responses to extracellular factors but also constitutes a potentially valuable source of various compounds. Plant growth regulators (PGRs) or phytohormones play an important role in determining the development pathway of plant cells and tissues in the culture medium. Auxins and cytokinins are the most widely used plant growth regulators in plant cell and tissue culture. These PGRs are well known to affect the physiological and morphological properties of cultured cells [24]. However, little is known about their effects on the composition of culture medium. Mori et al. [24] reported that CM is a conditioning factor to achieve cell growth and anthocyanin accumulation in strawberry cell suspension cultures, and Sakurai et al. [25] found that addition of CM from strawberry cell cultures to broth stimulated anthocyanin accumulation in suspension-cultured rose cells. Regarding mammalian cell cultures, the application of CM from mesenchymal stem cells was proposed as a new therapeutic strategy in regenerative medicine, as CM effected skin wound healing and tissue repair and showed anti-apoptotic, neuroprotective, and neurotrophic effects [26,27]. Recently, crude enzymatic extracts of CM obtained from *Botrytis cinerea* cultures were found to show bioconversion of resveratrol and pterostilbene to stilbene analogs including *δ*-viniferin [28]. So far, no studies have been conducted to assess bioconversion through CM from plant callus and cell suspension cultures; however, the use of CM of plant callus and cell suspension cultures may offer some advantages over the use of living microorganisms or enzymes, because CM can be harvested repeatedly from cell suspension cultures and can thus be easily separated from cells by filtration.

In this study, we investigated a new and simple method for the bioconversion of *δ*-viniferin from *trans*-resveratrol using the CM of grapevine callus suspension cultures. The results suggest that optimized bioconversion conditions of *trans*-resveratrol to *δ*-viniferin can achieve 99% bioconversion efficiency. CM harvested from grapevine callus suspension cultures on day 13 exhibited maximum bioconversion efficiency. To the best of our knowledge, this is the first study to report a simple method for bioconverting *trans*-resveratrol to *δ*-viniferin using CM from plant callus and cell suspension cultures.

## 2. Results and Discussion

### 2.1. Bioconversion of trans-Resveratrol to δ-Viniferin through CM of Grapevine Callus Suspension Cultures

Plant cells release numerous proteins, including peroxidases, to the CM during culturing [29,30,31,32], and we observed previously that overexpression of grapevine peroxidases in *Arabidopsis* plants led to the conversion of *trans*-resveratrol to *δ*-viniferin [33]. Thus, to investigate the bioconversion capacity of such CM, we performed in vitro bioconversion assays of *trans*-resveratrol using CM of grapevine callus suspension cultures (Figure 1A). HPLC analyses showed that the major peak was detected at the retention time (21.3 min) matching with *δ*-viniferin standard, representing that CM converted *trans*-resveratrol to *δ*-viniferin; however, this was only achieved through the addition of hydrogen peroxide (H_2_O_2_). The reaction in the absence of CM failed to convert *trans*-resveratrol to *δ*-viniferin (data not shown). Furthermore, the bioconverted *δ*-viniferin was identified by UPLC-Q-TOF MS spectrometry and nuclear magnetic resonance (NMR) spectroscopy. The bioconverted *δ*-viniferin exhibited ion peaks [M+H]^+^ at *m*/*z* 455.1 under ESI-positive mode (Appendix A). ^1^H-NMR and ^1^C-NMR spectra data also determined the compound to be *δ*-viniferin (Appendix A). *δ*-Viniferin production elicited a color change from transparent to yellow color in CM containing *trans*-resveratrol and H_2_O_2_, indicating bioconversion of *trans*-resveratrol to *δ*-viniferin (Figure 1B). The yellow color began to occur immediately after the addition of *trans*-resveratrol and H_2_O_2_ to the CM, suggesting that peroxidases were released from cells to the CM, which was likely involved in the oxidation of *trans*-resveratrol and the reduction of hydrogen peroxide (Figure 1C). CM of plant cell suspension cultures contains numerous plant peroxidases [31,34], and acidic and basic peroxidases were observed in CM of grapevine callus suspension cultures [23]. This suggests that certain secretory peroxidases secreted from cells into the CM during grapevine callus suspension cultures catalyzed the oxidative reaction of *trans*-resveratrol conversion to *δ*-viniferin by consuming hydrogen peroxide.

### 2.2. Effect of CM Obtained at Different Time Points of Callus Suspension Cultures on Bioconversion Efficiency

We examined peroxidase activity in CM according to the growth phases of grapevine callus suspension cultures. The growth curve of grapevine callus suspension cultures exhibited an initial lag phase during the first three days, followed by an exponential growth phase until day 9. After that, biomass yield decreased until day 13 (Figure 2A). As callus biomass gradually increased during culturing, we examined whether the activity of peroxidases secreted into the CM also increased. In vitro peroxidase activity assays using guaiacol as a substrate were carried out at two-day intervals for 13 days, and peroxidase activity in the CM increased markedly over time (Figure 2A) and was maximal at the end of the culture period (day 13).

To examine effects of culture time on bioconversion efficiency of the CM, we performed in vitro bioconversion assays of *trans*-resveratrol using CM obtained at different time points (Figure 2B). CM was collected at two-day intervals over 13 days, and bioconversion efficiency of *trans*-resveratrol to *δ*-viniferin increased from 26% to 61% with increasing culture time (Figure 2B). Bioconversion efficiency (61%) was highest using CM after 13 days of culturing, which was strongly correlated with peroxidase activity. These results suggested that peroxidases secreted into the CM during callus suspension cultures catalyzed the *trans*-resveratrol bioconversion. Similarly, previous studies reported extracellular peroxidase activities in plant cell suspension cultures of peanut, pepper, and bilberry cells, and peroxidase activity was highest at the end of the growth phase [35,36]. These studies demonstrated that an increase in peroxidase activity during cell suspension culturing resulted from either increased secretion of peroxidases during cell growth or higher stability of these enzymes compared to other extracellular proteins. Thus, secretion of peroxidases into the CM increased with increasing culturing time, which led to increased oxidation of guaiacol by extracellular peroxidases and increased bioconversion of *trans*-resveratrol to *δ*-viniferin.

We performed guaiacol peroxidase zymography using McLellan’s buffer (pH 4.4) to examine crude peroxidase activity in CM harvested at different time points of callus suspension cultures (Appendix A). Brown bands indicating peroxidase activity were observed upon incubating the native gel with guaiacol and H_2_O_2_. The band patterns on the zymogram indicated that two major basic peroxidase isozymes with at least one acidic enzyme were gradually activated over culture time with the highest activity on day 13. This was in line with the result of the guaiacol peroxidase activity assay using CM from 13-day cultures (Figure 2B). Additional bands with acidic peroxidase isozymes (located at bottom of the native gel) were also observed on the zymogram using CM from day 1 to day 11, but they were faint on day 13. This may have resulted from the low stability of acidic peroxidases during the stationary growth phase from day 11 to day 13.

### 2.3. Optimization of Bioconversion Conditions for trans-Resveratrol to δ-Viniferin

To optimize bioconversion conditions so as to obtain high yields of *δ*-viniferin using grapevine CM, the effects of several factors including incubation time, H_2_O_2_ concentration, temperature, and pH on bioconversion efficiency were examined. The effect of H_2_O_2_ concentration was tested using different concentrations (6.8, 14.1, and 21.9 mM) at room temperature, and bioconversion efficiency was highest at 6.8 mM H_2_O_2_ (43%), whereas it decreased slightly at 14.1 and 21.9 mM H_2_O_2_ (Figure 3A). To investigate the effect of incubation time, the assay was performed using different incubation times (i.e., 1, 5, 30, and 60 min; Figure 3B), and efficiency was highest (47%) after 30 min of incubation. With the addition of *trans*-resveratrol, incubation for 5 min showed 43% bioconversion efficiency, whereas after incubation for 1 min, it was reduced to 12% (Figure 3B). Moreover, bioconversion yield gradually increased with temperature increasing from 20 to 40 °C (Figure 3C). Bioconversion efficiency of approximately 60% occurred at 40–60 °C, and it markedly decreased to 16% at 80 °C. Maximum bioconversion efficiency of 60% occurred at 60 °C. Considering that the optimum temperature of phenol oxidation of free horseradish peroxidases is 45 °C and its activity decreases to 19% at 50 °C [37], extracellular peroxidases in grapevine CM may be highly thermostable.

We further investigated the effect of pH in a range from 3.0 to 8.0, and bioconversion efficiency was highest at pH 6.0 (64%) under the optimized conditions of 60 °C, 30 min incubation time, and 6.8 mM H_2_O_2_. At pH 5.0–7.0, efficiency was at least 50%; however, it decreased to 32% at pH 8.0 and to 11% at pH 3.0 (Figure 3D). These results suggested high bioconversion efficiency by extracellular peroxidases in grapevine CM under acidic conditions (pH 5.0–6.0) at 40–60 °C with 30–60 min incubation time and with 6.8 mM H_2_O_2_.

### 2.4. Bioconversion of trans-Resveratrol through CM from Different Callus Suspension Cultures

Extracellular peroxidases are secreted to culture medium in plant cell suspension cultures [31]; thus, we examined bioconversion of *trans*-resveratrol using CM from different callus suspension cultures (Figure 4). Among seven plant callus suspension cultures used in this study, CM from five cultures, i.e., *Dianthus chinensis* (27%), *Aronia melanocarpa* (48%), *Glycine max* (44%), *Lonicera japonica* (33%), and *Zanthoxylum schinifolium* (50%), exhibited more than 30% of bioconversion efficiency, whereas CM from two cultures, i.e., *Sesamum indicum* (5%) and *Taraxacum coreanum* (1%), showed less than 10% bioconversion efficiency. CM from three callus suspension cultures, i.e., *A. melanocarpa* (48%), *Z. schinifolium* (50%), and *G. max* (44%), exhibited higher bioconversion efficiency than that of *V. labruscana* (43%). We also measured in vitro peroxidase activity using CM from callus suspension cultures (Figure 4B). Peroxidase activities displayed similar patterns to the bioconversion efficiency. The highest peroxidase activity was observed in CM from *G. max*. CM from *Z. schinifolium* and *G. max* showed higher peroxidase activity than that of *V. labruscana* cultures. CM from two cultures (*S. indicum* and *T. coreanum*) displayed very low peroxidase activity, in line with their low bioconversion efficiency (Figure 4A). The correlation of bioconversion efficiencies and peroxidase activities suggests that CM from most callus suspension cultures contained extracellular peroxidases, thereby facilitating bioconversion of *trans*-resveratrol to *δ*-viniferin in vitro. Our results thus demonstrate that CM of plant callus suspension cultures may be used as a simple and low-cost method to bioconvert *trans*-resveratrol to *δ*-viniferin.

## 3. Materials and Methods

### 3.1. Plant Materials and Callus Culture Conditions

Callus derived from grapevine (*V**. labruscana* L. cv. Cambell Early, BP1347372), *Dianthus chinensis* (BP1347320), *Lonicera japonica* (BP1421766), *Aronia melanocarpa* (BP1347380), *Zanthoxylum schinifolium* (BP1421800), *Glycine max* (BP1421820), *Sesamum indicum* (BP1345716), and *Taraxacum coreanum* (BP1347324) was distributed from the Korean Collection for Type Cultures (KCTC). The induced calli were maintained on MS1D solid medium (Murashige and Skoog basal medium including vitamins, supplemented with 1 mg L^−^¹ 2,4-D, 30 g L^−^¹ sucrose, 0.5 mg L^−^¹ MES, and 0.4% gelrite) in the dark as previously described [15,33]. Calli were subcultured every three weeks by transferring healthy and homogeneous calli to MS1D solid medium.

### 3.2. Preparation of CM from Plant Callus Suspension Cultures

Callus suspension cultures were produced by transferring healthy and homogeneous calli (approximately 2 g fresh weight, 10% (*v*/*v*) inoculum) into a 125-mL flask containing 20 mL MS1D liquid medium. The flasks were placed on a rotary shaker at 90 rpm in the dark at 24 °C for five days. CM was separated by filtration of callus culture medium using a nylon mesh filter (100 µm) and was further purified by centrifugation at 9000× *g* for 10 min. CM was harvested from grapevine callus suspension cultures at two-day intervals over 13 days after inoculation of grapevine calli.

### 3.3. Bioconversion of trans-Resveratrol Using CM

Bioconversion of trans-resveratrol was carried out using CM from plant callus suspension cultures. CM (0.6 mL) was added to a 1.5-mL tube containing 0.3 mL of water, and aqueous hydrogen peroxide (6.8 mM H_2_O_2_; 2 µL 10% H_2_O_2_) was added. The mixture was incubated at 25 °C on an orbital shaker at 100 rpm for 1 min. A mixture of trans-resveratrol (1 mg) dissolved in 0.1 mL 100% acetone was added, and the solution was further incubated at 25 °C on an orbital shaker at 100 rpm for 5 min.

### 3.4. δ-Viniferin Analysis and Quantification

The resulting reaction solution was extracted using liquid/liquid extraction with ethyl acetate. The ethyl acetate layer was then air-dried, diluted with 80% methanol, and filtered using a 0.2-μm hydrophilic PTFE membrane filter (Advantec MFS, Inc., Dublin, CA, USA). The samples were analyzed through HPLC as previously described [15]. *trans*-Resveratrol and *δ*-viniferin were identified by comparison with a commercial standard (Sigma-Aldrich, St. Louis, MO, USA) and with *δ*-viniferin as reported in our previous study [15], respectively. The *δ*-viniferin bioconverted by CM was analyzed through UPLC-Q-TOF MS using the Viron UPLC^TM^ system (Viron, Waters, Milford, MA, USA). The LC condition was optimized on an Acquity UPLC BEH C18 column (2.1 mm × 100 mm, 1.7 μm; Waters, Milford, MA, USA). The flow rate was 0.35 mL/min, and mobile phases included solvent A water with 0.1% formic acid (FA), and solvent B acetonitrile (ACN) with 0.1% FA. The MS operating conditions were as follows: sample cone voltage 30 V, capillary voltage 3.0 kV, ion source temperature 100 °C, cone gas 30 L/h, desolvation gas 800 L/h, desolvation temperature 400 °C, scan range *m*/*z* 50–1500, scan time 0.2 s, and collision energy ramp from 10 to 30 eV (*m*/*z* 50–1500). The spectrometer was used with electrospray ionization (ESI) in positive modes. The structure of bioconverted *δ*-viniferin was confirmed by NMR. NMR spectra were recorded on a JEOL (JNM-ECZ500R) instrument (^1^H NMR at 500 MHz, ^13^C NMR at 125 MHz; JEOL, Tokyo, Japan) in methanol-d4.

The amounts of bioconverted *δ*-viniferin were calculated based on a standard curve of *δ*-viniferin. Bioconversion efficiency was defined as the ratio of the amount of *δ*-viniferin bioconverted from *trans*-resveratrol. The bioconversion efficiency of *trans*-resveratrol to *δ*-viniferin was calculated using the following formula:Efficiency (%) = 100 × (mg of bioconverted *δ*-viniferin/1 mg of *trans*-resveratrol)

### 3.5. Optimization of trans-Resveratrol Bioconversion

To identify the optimal conditions for bioconversion through grapevine CM, the effects of several factors on bioconversion conditions were measured using CM harvested after five days of grapevine callus suspension culturing. The effects of incubation time (1, 5, and 60 min) and H_2_O_2_ concentration (6.8, 14.1, and 21.9 mM) on bioconversion efficiency were tested. The influence of temperature on *trans*-resveratrol bioconversion was carried out by using CM at different temperatures (20, 30, 40, 50, and 60 °C). To investigate the effect of pH on *trans*-resveratrol bioconversion efficiency, CM was used at pH 3.0, 4.0, 5.0, 6.0, 7.0, and 8.0. The pH of CM was adjusted by adding a small amount of 0.2 M citric acid (pH 1.8), 0.2 M Na_2_HPO_4_ (pH 8.5), or Tris-HCl (pH 8.0). Each experiment was replicated three times.

### 3.6. Crude Peroxidase Activity Assay and Native Polyacrylamide Gel Electrophoresis (PAGE)

Total crude peroxidase activity in CM was determined using the guaiacol oxidation method in the presence of hydrogen peroxide and with a reaction time of 5 min. The reaction volume (1 mL final volume) contained 0.1 mL CM and 0.9 mL of the reaction mixture containing 50 mM phosphate-buffered saline (PBS; pH 7.0), 25 mM guaiacol, and 25 mM H_2_O_2_. Guaiacol oxidation was measured by absorbance at 470 nm using a Biochrom Ultraspec 2100 pro UV/Vis spectrophotometer (Thermo Fisher Scientific, Waltham, MA, USA). Crude peroxidase activities during different incubation times were measured as the increase in absorbance, with the same reaction mixture and assay conditions. Crude peroxidase activity in CM was recorded as the mean of three replicates. The peroxidase activity in the CM was calculated using the crude peroxidase activity of the conditioned medium, calculated using the formula in the equation as described previously [38].
$$\frac{U}{\mathrm{mL}}=\frac{\Delta \mathrm{OD}}{\min}\times \frac{\mathrm{RmV}}{\mathrm{EV}}\times \frac{\mathrm{dF}}{\varepsilon 470}$$

ΔOD/min = increase in absorbance per minute (min^−1^)RmV = reaction mixture volume (mL)EV = enzyme extract volumedF = dilution factorε470 = molar absorptivity of tetraguaiacol at 470 nm (mL μ^−1^ cm^−1^)

Crude peroxidase activity in CM was assessed through PAGE as previously described [33]. Briefly, 30 μL non-boiled CM was dissolved in 5X sample buffer on a 12% native PAGE gel (pH 4.4) and was electrophoresed at 100 V for 180 min at 4 °C using a protein gel electrophoresis chamber system (Hoefer, San Francisco, CA, USA). After electrophoresis, the gel was incubated in a guaiacol reaction mixture (50 mM PBS, pH 7.0, 25 mM guaiacol, and 25 mM H_2_O_2_) until peroxidase activity bands became visible.

### 3.7. Statistical Analyses

All experimental data were subjected to a one-way analysis of variance and Duncan’s post hoc tests using Prism (GraphPad software, v 5. 03; GraphPad Software, San Diego, CA, USA). Statistical significance was reported at *p* ≤ 0.05.

## Figures and Tables

**Figure 1 ijms-23-04403-f001:**
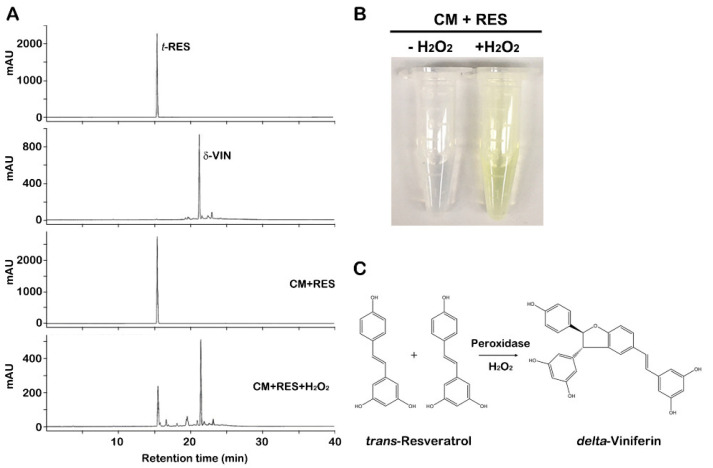
Bioconversion of *trans*-resveratrol (RES) using the conditioned medium (CM) from grapevine callus suspension cultures. (**A**) HPLC chromatograms of bioconversion of *trans*-resveratrol to *δ*-viniferin using the grapevine CM. *trans*-Resveratrol was effectively transformed to *δ*-viniferin in the presence of hydrogen peroxide (H_2_O_2_). (**B**) Color changes of the CM in the presence of H_2_O_2_. The bioconversion experiment was carried out using the CM with *trans*-resveratrol in the absence of H_2_O_2_ (**left**) or the presence of H_2_O_2_ (**right**). (**C**) Proposed mechanism for the formation of *δ*-viniferin from *trans*-resveratrol catalyzed by extracellular peroxidases present in CM.

**Figure 2 ijms-23-04403-f002:**
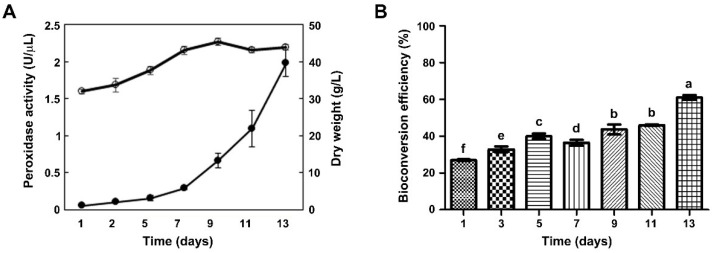
Effect of culturing time on peroxidase activity in the CM from grapevine callus suspension cultures. (**A**) Guaiacol peroxidase activity in the CM (filled dots) and cell growth curve (biomass, dry weight) of grapevine callus suspension cultures (open dots) were measured at two-day intervals for 13 days of culturing. (**B**) The bioconversion efficiency of *trans*-resveratrol to *δ*-viniferin was measured using the grapevine CM with different culture periods. Grapevine callus (2 g) was cultured in 20 mL MS1D liquid medium, and the CM supernatant was harvested at the indicated time points after callus suspension culturing. The bioconversion was conducted by adding *trans*-resveratrol (1 mg) and H_2_O_2_ (6.8 mM) to the CM. Data are the means of three independent replicates ± SD. Same letter(s) indicate no significant difference (Duncan’s test; *p* ≤ 0.05).

**Figure 3 ijms-23-04403-f003:**
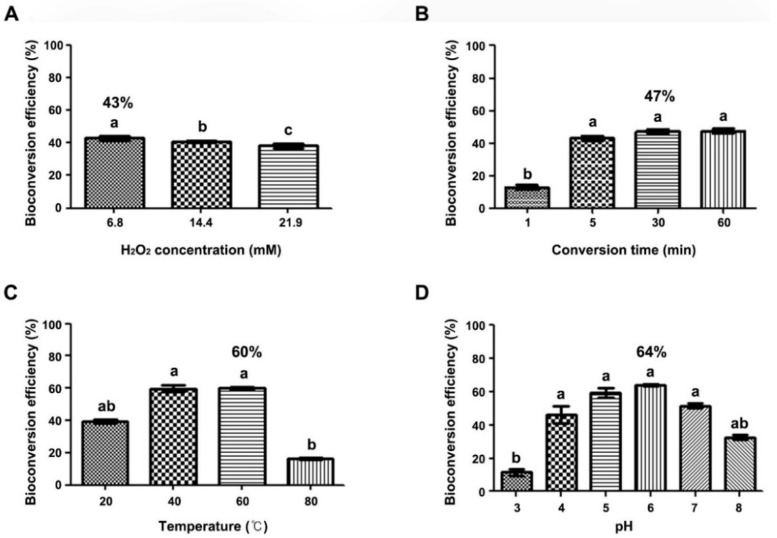
Optimization of reaction conditions for efficient bioconversion of *δ*-viniferin production from *trans*-resveratrol using the grapevine CM. Grapevine callus (2 g) was cultured in 20 mL MS1D liquid medium and the CM supernatant was harvested at the indicated time points after callus suspension cultures. The bioconversion was conducted by adding *trans*-resveratrol (1 mg) and H_2_O_2_ (6.8–21.9 mM) into the CM. (**A**) Hydrogen peroxide concentration, (**B**) incubation time, (**C**) temperature, and (**D**) pH. Shown are the means of three independent replicates ± SD. Same letter(s) indicate no significant difference (Duncan’s test; *p* ≤ 0.05).

**Figure 4 ijms-23-04403-f004:**
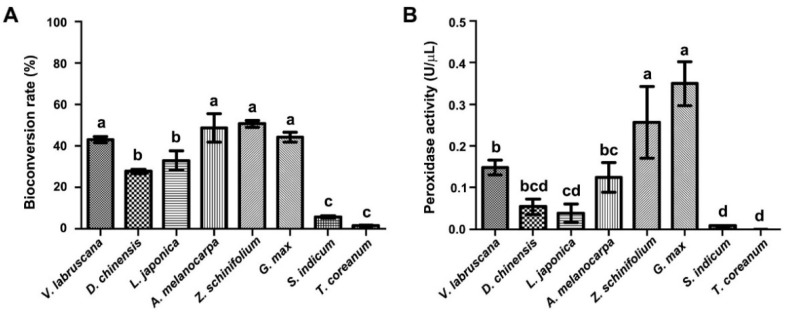
Determination of bioconversion efficiency (**A**) and guaiacol peroxidase activity (**B**) using the CM from various plant callus suspension cultures. The various plant calli (2 g) were cultured in 20 mL MS1D liquid medium, and CM was collected after five days of callus suspension culturing. The guaiacol peroxidase activity was determined by the reaction of 0.1 mL CM and 0.9 mL of the reaction mixture (25 mM guaiacol, 25 mM H_2_O_2,_ 50 mM phosphate-buffered saline, pH 7.0) within 5 min. Values are presented as means ± SD. Same letter(s) indicate no significant difference (Duncan’s test; *p* ≤ 0.05).

## Data Availability

All datasets for this study are included in the manuscript and the Appendix A.

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
