# Peer review of "Highly Efficient Bioconversion of trans-Resveratrol to δ-Viniferin Using Conditioned Medium of Grapevine Callus Suspension Cultures"

_ijms, 2022, doi:10.3390/ijms23084403_

Round 1

Reviewer 1 Report

I find that manusctipt entiteled "Highly Efficient Bioconversion of trans-Resveratrol to δ-Viniferin Using Conditioned Medium of Grapevine Callus Suspension Cultures" is acceptable for publication. It presents an alternative method for bioconversion of trans-resveratrol. All comments of the editor were addressed and additional explanations given and corrections and improvements made in the manuscript.  

Reviewer 2 Report

The manuscript has been improved based on the reviewer’s comments. However, my comments (questions) mentioned in the previous review are not answered. I still have some questions about this paper.

Q1; Is there another negative control reaction for Fig. 1A and B? For example, a reaction of trans-resveratrol just with H2O2, without CM. “RES+H2O2” reaction is necessary to verify the effect of CM.

Q2; Phytohormones (plant hormones; auxins or cytokinins) affect the physiological property of cultured cells and also the CM (supernatant medium) composition probably. However, any information or example explanation about the effects of the composition of the culture medium is not presented. In the introduction or discussion section, the effects of the culture medium composition, especially 2,4-D, should be referred to or discussed.

Round 2

Reviewer 2 Report

The author has replied to the reviewer's comments. The manuscript has been revised based on that.

This manuscript is a resubmission of an earlier submission. The following is a list of the peer review reports and author responses from that submission.

Round 1

Reviewer 1 Report

This manuscript demonstrates the bioconversion (artificial conversion) of trans-resveratrol to δ-viniferin in vitro by using the filtrate of grapevine callus suspension-cultured media. The callus suspension culture media called “Conditioned Medium” were prepared from eight plant species including grapevine. Although two of those were low efficiency, the CMs of Vitis labrusca, Aronia melanocarpa, Zanthoxylum schinifolium, and Glycine max showed high activity to convert trans-resveratrol to δ-viniferin (Fig. 4A). The conversion efficiency seems to be corresponded to peroxidase activity in the CM (Fig. 2 and Fig. 4). The efficient condition of reaction for bioconversion were assayed using H2O2 and different temperature and pH conditions (Fig. 1 and Fig. 3). The experimental design and the description of the results are comprehensible. I think this manuscript is almost OK to accept.

I have a few questions or suggestions for further improvement of this manuscript.

Q1; Is there another negative control, for example, a reaction of trans-resveratrol just with H2O2, without CM, for Fig. 1?  I would like to know the result of the negative control reaction  “RES+H2O2” in Fig. 1.

Q2; The definition or calculation method for peroxidase activity (U/µL) is unclear from the description in the materials and methods section. How was the unit (U) defined?

Q3; The calculation method of bioconversion efficiency is also unclear from the description in the materials and methods section. How was the efficiency(%) was calculated?

Q4; Why is 2,4-D used for the callus culture medium? Are there any references (previous studies) to compare different plant growth regulators to prepare the CMs? Is 2,4-D the most effective PGR to produce secreted proteins from the callus? Please add some references to explain the MS1D medium.

Minor errors

P.1 L.19; Vitis. labruscana >> Vitis labruscana

Author Response

Comment 1:

Is there another negative control, for example, a reaction of trans-resveratrol just with H2O2, without CM, for Fig. 1?  I would like to know the result of the negative control reaction “RES+H2O2” in Fig. 1.

â–¶ Response:

As the reviewer’s comment, we examined the bioconversion of trans-resveratrol with MS1D liquid media and H2O2 as another negative control. The reaction was not able to convert trans-resveratrol to δ-viniferin.

Comment 2:

The definition or calculation method for peroxidase activity (U/µL) is unclear from the description in the materials and methods section. How was the unit (U) defined?

â–¶ Response:

As the reviewer’s comment, we addressed description of the unit (U) in the section of Materials and methods as follows (page 9);

The crude peroxidase activity of the conditioned medium has been calculated using the formula in the equation as described previously (Ambatkar et al. 2013).

Equation:

ΔOD/min = increase in absorbance per minute (min-1)

RmV = reaction mixture volume (mL)

EV = enzyme extract volume

dF = dilution factor

ε470 = molar absorptivity of tetraguaiacol at 470 nm (mL m-1 cm-1)

One unit (U) is the amount of enzyme that converts 1 mole of the substrate into a product per minute.

Comment 3:

The calculation method of bioconversion efficiency is also unclear from the description in the materials and methods section. How was the efficiency (%) calculated?

 â–¶ Response:

As the reviewer’s comment, we described the calculation of the bioconversion efficiency (%) in the section of Materials and methods as follows (page 8);

The bioconversion efficiency of trans-resveratrol to δ-viniferin was calculated using the following formula:

Efficiency (%) = 100  (mg of bioconverted δ-viniferin / 1 mg of trans-resveratrol)

Comment 4:

Why is 2,4-D used for the callus culture medium? Are there any references (previous studies) to compare different plant growth regulators to prepare the CMs? Is 2,4-D the most effective PGR to produce secreted proteins from the callus? Please add some references to explain the MS1D medium.

â–¶ Response:

2,4-D is widely used in plant cell cultures for callus induction and maintenance in most plant species. We previously used 2,4-D as plant growth regulator to induce callus from grapevine (Jeong et al., 2020, Park et al., 2020). As the reviewer’s comment, we have added references to explain the MS1D medium in the section of Materials and methods as follows (page 8);

The induced calli were maintained on MS1D solid medium (Murashige and Skoog basal medium including vitamins, supplemented with 30 g L-¹ sucrose, 0.5 mg L-¹ MES, 1 mg L-¹ 2,4-D, and 0.4% gelrite) in the dark as previously described (Jeong et al., 2020, Park et al., 2020).

Reviewer 2 Report

Manuscript entitled "Highly Efficient Bioconversion of trans-Resveratrol to δ-Viniferin Using Conditioned Medium of Grapevine Callus Suspension Cultures" presents interesting results obtained trough well performed research using appropriate methodology.
I find this manuscript interesting to the readers and acceptable for publication in present form.
One advice for future researches is to apply experimental design for optimisation of bioconversion conditions rather than one parameter at the time approach.
